# Factor XII-Deficient Chicken Plasma as a Useful Target for Screening of Pro- and Anticoagulant Animal Venom Toxins

**DOI:** 10.3390/toxins12020079

**Published:** 2020-01-23

**Authors:** Benedito C. Prezoto, Nancy Oguiura

**Affiliations:** 1Laboratory of Pharmacology, The Butantan Institute, Av. Dr. Vital Brazil 1500, São Paulo CEP 05503-900, Brazil; 2Ecology and Evolution Laboratory, The Butantan Institute, Av. Dr. Vital Brazil 1500, São Paulo CEP 05503-900, Brazil; nancy.oguiura@butantan.gov.br

**Keywords:** thromboelastimetry, phospholipase, snake venom, insect venom

## Abstract

The sensitivity of vertebrate citrated plasma to pro- and anticoagulant venom or toxins occurs on a microscale level (micrograms). Although it improves responses to agonists, recalcification triggers a relatively fast thrombin formation process in mammalian plasma. As it has a natural factor XII deficiency, the recalcification time (RT) of chicken plasma (CP) is comparatively long [≥ 1800 seconds (s)]. Our objective was to compare the ability of bee venom phospholipase A_2_ (bvPLA_2_) to neutralize clot formation induced by an activator of coagulation (the aPTT clot) in recalcified human and chicken plasmas, through rotational thromboelastometry. The strategy used in this study was to find doses of bvPLA_2_ that were sufficient enough to prolong the clotting time (CT) of these activated plasmas to values within their normal RT range. The CT of CP was prolonged in a dose-dependent manner by bvPLA_2_, with 17 ± 2.8 ng (n = 6) being sufficient to displace the CT values of the activated samples to ≥ 1800 s. Only amounts up to 380 ± 41 ng (n = 6) of bvPLA_2_ induced the same effect in activated human plasma samples. In conclusion, the high sensitivity of CP to agonists and rotational thromboelastometry could be useful. For example, during screening procedures for assaying the effects of toxins in several stages of the coagulation pathway, such as clot initiation, formation, stability, strength, or dissolution.

## 1. Introduction

Studies of animal venoms and toxins have focused on one or more of the following objectives: (i) To determine the mode and mechanism of action of the toxins; (ii) to find ways and means to neutralize the toxicity and adverse effects of accidents; (iii) to develop specific research tools that are useful in understanding the normal physiological processes at both cellular and molecular levels; (iv) to develop prototypes of pharmaceutical agents based on the structure of toxins [1]. 

Tests to assess the in vivo effects of animal venoms, such as hemorrhage, myonecrosis, defibrination, edema, and venom lethality/antivenom potencies, still largely rely on animal models (usually involving rodents). However, these assays lead to animal suffering. Stringent regulations governing the use of animals, limited research funds, and public pressure all drive the need for the development of alternative in vitro assays involving non-animal, or nonsentient research methods, as a way to achieve the ‘Three Rs’ goals of animal experimentation, i.e., Reduce, Refine, and Replace animal tests. It is widely known that animal venoms contain enzymes and non-enzymatic proteins that interfere with hemostasis leading to hemorrhage or even thrombosis. Regarding hemotoxins, current scientific attention is focused on the validation of alternative methods to the screening and characterization of individual venom proteins, which should, in turn, enable novel in vitro assays to be designed, thus reducing the number of animals required [2]. 

Technologies that show a good correlation coefficient between the venom pro- and anticoagulant effects on mammalian plasmas in vitro and lethality in vivo have been reported [3,4,5,6,7,8,9,10,11,12,13,14,15]. However, these techniques present some limitations: (i) Many studies simply characterize the toxin conversion of isolated substrates, giving a single parameter for one complex enzymatic process. As the full effects of toxins on the coagulation pathway are rarely examined, even in vitro, our understanding of the pathophysiology of envenoming is limited [16] and (ii) non-recalcified plasma samples, such as those used in the minimum coagulant dose assay require relatively large amounts of venom (on the µg or mg scale) [17]. If the potentiation of action of agonists could be achieved by the addition of the cofactors Ca^2+^ and phospholipids [18], this strategy could accelerate some enzymatic reactions in the coagulation cascade [19]. As a consequence, the time interval in which agonists can be assayed becomes limited [to around 600 seconds (s)]. This limitation associated with mammalian plasma becomes evident during screening strategies, when small amounts of the proteins being tested are available, for example, during the purification procedures of individual toxins from crude venoms. 

Four constituents form the mammalian intrinsic pathway of coagulation: The trypsin-type serine proteases factors XIIa, XIa, plasma kallikrein and the cofactor high molecular weight kininogen [20]. Extreme slowness in spontaneous in vitro thrombin/fibrin generation even after the addition of Ca^2+^ ions is a hallmark of the plasma of patients with a factor XII deficiency [21]. The main difference between the human and chicken coagulation process is that the factor XII gene is completely missing in the chicken genome [22]. Rotational thromboelastometry improves the evaluation of the clotting process, since it monitors several parameters, such as the stages of clot initiation, formation, stability, strength, and dissolution [23]. This technology has been used in various studies reporting on the pro- and anticoagulant activities of several snake venoms on citrated human whole blood, as well as plasma samples of rats and dogs [24,25,26,27,28,29]. By testing recalcified chicken plasma (CP) samples through rotational thromboelastometry, we recently published two studies in which we describe the abilities of the bothropic and crotalic antiserums in neutralizing the pro- and anticoagulant activities of *Bothrops jararaca* (*B. jararaca*) venom and crotoxin, respectively. We found a positive correlation between the data obtained with these in vitro assays and data from the in vivo lethality technique related to the *B. jararaca* and *Crotalus durissus terrificus* (*C. d. terrificus*) venoms [30,31]. Predictably, addition of the cofactors Ca^2+^ and phospholipids factor XII-deficient CP elicited a time lapse sufficient for the elaboration of one typical dose–response curve, establishing its sensitivity to these agonists and to antagonists in the nanoscale range.

Phospolipase A_2_ (PLA_2_s) (EC 3.1.1.4) are a large family of proteins found in various mammalian tissues and in the venoms of snakes and arthropods. The toxins that are predominately responsible for the neurotoxic effects of snake venoms are members of the diverse PLA_2_ and three-finger toxin (3FTX) families [32,33,34]. One important pharmacological effect of certain secreted PLA_2_ toxins is the in vitro anticoagulant activity on human plasma samples [35,36,37]. The bee (*Apis mellifera*) (*A. mellifera*) venom resulted in no in vitro coagulant activity toward human plasma or fibrinogen, but had an anticoagulant effect. It did not destroy fibrinogen, inactivate thrombin, nor interfere with the interaction between thrombin and fibrinogen. Hence, its anticoagulant effect does not appear to occur in the final step of blood coagulation (fibrinogen to fibrin transformation or proteolytic action of thrombin). The anticoagulant action of this venom appears to be due to the inhibition of prothrombin activation, while tissue thromboplastin, cephalin, and ruptured platelets were able to counteract the anticoagulant effect of the venom [38]. Bee venom PLA_2_ (bvPLA_2_) and melittin are the most important and abundant muscle-damaging components and key factors implicated in the pathophysiology of envenomations induced by bee *A. mellifera* venom [39,40]. Recently, Nielsen reported consistent anticoagulant activity of bvPLA_2_ when tested on recalcified and nonactivated normal human plasma samples, at a concentration of 200 ng/mL [41]. 

The unusually high sensitivity of recalcified CP to the pro- and anticoagulant effects of *B. jararaca* venom and crotoxin, respectively, encouraged us to compare the sensitivities of recalcified chicken and human plasma previously activated with a standardized doses of an ellagic acid- and phospholipid-based reagent (aPTT clot) in relation to the in vitro anticoagulant activity of bvPLA_2_, under similar conditions, through rotational thromboelastometry. 

## 2. Results

### 2.1. Standardization of the Activator aPTT Clot Mean Coagulant Dose on Chicken and Human Plasma Samples

The clotting time (CT) values of the control-treated samples of recalcified CP [(corresponding to the recalcification time (RT)] were significantly greater (mean = 2247 ± 319 s, n = 6) than those presented by human plasma (874 ± 230 s, n = 6). The addition of 0.6, 6, and 60 µL of the aPTT clot reagent induced CT values of 1775 ± 323, 1262 ± 282, and 279 ± 125 s, respectively (n = 6 in each experimental group), in the CP samples (Figure 1). For the elaboration of the dose–response curve using CP, reference values of 90, 900, and 1800 s were considered as the maximum, mean, and minimum coagulant responses, respectively, of the plasma to the tested doses of the activator aPTT clot reagent. Doses between 6 and 18 µL (inducing to a CT value of 900 s) were then defined as the mean coagulant dose (MCD) of the activator aPTT clot reagent for each particular assay with CP. For human plasma, the addition of 0.6, 6, and 60 µL of the aPTT clot reagent induced CT values of 454 ± 102, 347 ± 95, and 149 ± 60 s, respectively (n = 6 each experimental group) (Figure 2). Doses between 5 and 9 µL [placing the CT parameter in an interval of 100 to 240 s, the normal range values for the intrinsic pathway thromboelastometry (INTEM) assay using human plasma] were then defined as the MCD of the activator aPTT clot reagent for each particular assay of the experimental groups with human plasma. 

### 2.2. Standardization of the Effective Anticoagulant Dose of the Bee A. mellifera Venom Phospholipase A_2_ on Chicken and Human Recalcified and Activated Plasma Samples

BvPLA_2_ showed anticoagulant activity in a dose-dependent manner, when added to CP that was simultaneously recalcified and activated with the MCD of the activator aPTT clot reagent; 17 ± 2.8 ng and 9 ± 2.2 ng were considered as its effective and mean anticoagulant doses (EAD and MAD), respectively (Figure 3; Figure 4). On the other hand, the EAD of bvPLA_2_ on human plasma was of 380 ± 41 ng (n = 6) (Figure 5). 

## 3. Discussion 

The relatively large RT presented by CP allowed for the elaboration of a dose–response curve, and the determination of the relative potencies of both the agonist (aPTT clot reagent) and antagonist (bvPLA_2_), which caused its sensitivity to be almost 20-fold higher than that of human plasma. Nielsen recently showed through thromboelastography that effective anticoagulant activity on normal human plasma was achieved by using bvPLA_2_ at 200 ng/mL [41]. In our conditions, 380 ± 41 ng was sufficient to displace the CT values of human plasma to values within the time intervals of their related RT in 100% of the assays. This discordance in results is likely due to this author using nonactivated human plasma samples. 

In summary, three clear differences can be observed in our study, when the INTEM profiles of activated CP samples are compared with that of human plasma: (i) The CT parameter values of the control recalcified CP is comparatively greater (mean = 2247 ± 319 s versus 874 ± 230 s); (ii) a statistically significant difference between the minimum and maximal coagulant responses to the activator aPTT clot reagent occur within two orders of magnitude (from 0.6 to 60 µL); and (iii) the bvPLA_2_ anticoagulant effect occurs in a dose-dependent manner. This unusual sensitivity of activated and recalcified CP to the bvPLA_2_ anticoagulant effect described herein reinforces our previous studies with this pharmacological target [30,31]. 

Below, we present arguments justifying why, in our opinion, platelet-poor CP samples should be considered in some studies using rotational thromboelastometry, for screening purposes of some animal venom toxins that interfere with several steps of the clotting process, such as the stages of clot initiation, formation, stability, strength, and dissolution. 

The careful collection and centrifugation of blood samples from chickens is required. Foam or bubble formation in the syringes during blood sample collection leads to cell lysis and possible activation of the coagulation process, shortening the RT of the control-treated CP samples. Some parameters of the ROTEM profile, such as the clotting time, clot formation time, α angle and maximal amplitude, are particularly dependent on fibrin polymerization and platelet count [42,43,44]. Recalcified CP samples obtained after centrifugation at greater rates render no typical ROTEM profile, probably as a consequence of low concentrations of cofactors, such as, for example, platelet microparticles and phospholipids [45,46]. To obtain one typical ROTEM profile, such as that shown in Figure 4, white Leghorn whole blood samples must be centrifuged at lower rates (up to 4000× *g*). Vehicle-treated recalcified CP samples are stable (no spontaneous clot formation) for at least 30 min. This prolonged RT is essential for our main purpose: Elaboration of one sensitive dose-response curve to agonists. Pilot assays with chicken whole blood samples or platelet-rich plasma present RT values of 705 ± 338 s and 1045 ± 428 s, respectively (n = 8, each) (data not shown). These restricted intervals become impracticable for the construction of one dose-response curve to these agonists. Consequently, platelet-rich plasma or whole blood samples presented sensitivities that were significantly smaller to the bvPLA_2_ anticoagulant activity, when compared with that of platelet-poor plasma samples.

The time taken for spontaneous clot formation following recalcification is highly variable between the plasma of mammal and nonmammal species. This may be due to a variation in the level of factor XII (Hageman factor) present [47]. Although it has a natural factor XII deficiency [22], CP possesses a fully functional extrinsic pathway, and its key coagulant proteins (factors V and X) could be considered as suitable targets or substrates for both procoagulant [30,48,49] and anticoagulant toxins [31] from several animal venoms. 

Our findings indicate that RT can be considered as the main difference between chicken and human plasma, in the conditions herein described. The relatively limited interval (almost 600 seconds) of RT and the relatively large reference intervals presented by the values of the CT parameter of the INTEM profile (from 100 to 240 s) in activated human plasma abrogates the possibility of the elaboration of a true dose–response relationship for the agonists here described. The differences between the CT values related to 0.6, 6 and 60 µL of the aPTT clot reagent were not statistically significant (Figure 2). On the other hand, the relatively large RT presented by CP elicited the elaboration of one dose–response curve, and the determination of the relative potencies of both the activator aPTT clot reagent and bvPLA_2_. 

## 4. Conclusions

The main limitations of most in vitro techniques designed for assaying pro- and anticoagulant activity of animal venoms and toxins on mammalian plasma is that these assays give a single parameter (fibrin formation) for one complex enzymatic process [16]. This limitation becomes evident, for example, during screening procedures of animal venoms, when the effects of toxins, such as fibrinolysis, fibrinogenolysis or fibrinolysis induction on the coagulation pathway should be examined. In conclusion, we propose this functional assay as an alternative for (i) screening assays during the purification of fibrinolytics, pro- or anticoagulant toxins from whole animal venoms, (ii) assessing a specific antivenom’s relative potencies against venom toxic components with both coagulant and lethal activities (such as *B. jararaca* venom [30] or with PLA_2_-like substances, such as those present in the venoms of the *C. d. terrificus* snake [31] or the *A. mellifera* bee, and (iii) studies for testing the relative potencies of prototypes of pharmaceutical agents or natural inhibitors of PLA_2_-like toxins from animal venoms in more diluted solutions, thus with less potential to affect the normal coagulation kinetic behavior.

## 5. Material and Methods

### 5.1. Reagents

An activated partial thromboplastin time reagent (aPTT) clot, containing ellagic acid and synthetic phospholipids, was obtained from BIOS Diagnostica (SP, Brazil); sodium citrate was obtained from Ecibra (Curitiba, Brazil); calcium chloride was obtained from E. Merck (German); pooled 4% citrated normal human plasma (maintained at −80 °C) and PLA_2_, from *A. mellifera* venom, were obtained from Sigma-Aldrich, St. Louis, MO, USA). PLA_2_ was dissolved in phosphate-buffered saline (PBS) at a concentration of 1 mg/ml, aliquoted for single-use, and frozen at −80 °C. The tested enzymes were only frozen and thawed once for all subsequently described work. All chemicals were of an analytical reagent grade.

### 5.2. Animals

Adult female or male white Leghorn chickens (1.0 to 1.7 kg) were used. All birds were a donation from commercial breeding (Granja Ino, São Paulo, Brazil). The animals had free access to water and food, and were kept in a 12 h light/dark cycle. 

Ethical approval: All the procedures involving animals were carried out following the Guiding Principles for the Use of Animals in Toxicology (International Society of Toxicology, http://www.toxicology.org) and the Brazilian College of Animal Experimentation (COBEA). The experimental protocol was approved by the Ethic Committee on Animal Use of the Butantan Institute (CEUAIB) (protocol number CEUA 6259250918) in 11/08/2018. 

### 5.3. Collection of Citrated Chicken Plasma Samples

The birds were restrained on their backs with their wings spread. After the use of xylocaine spray as a local anesthetic agent, the feathers were removed, and small incisions were made for cleaning around the brachial wing vein. Eight milliliters of whole blood samples were collected using syringes containing 1:10 (v/v) 3.2% trisodium citrate and then closed with cotton-yarn. Chicken plasma was obtained after centrifugation at 3000× *g* for 20 min at 4 °C. Plasma samples were used immediately or maintained at −80 °C. The total blood volume of one particular organism is very difficult to determine and depends on its species, sex, age, and health, as well as its nutritional condition. The total circulating blood volume is in the range 55–70 ml/kg of the body weight, and the adult chickens provided at least 8 mL of whole blood samples from each wing vein, without significant animal distress or the need for euthanasia.

### 5.4. Thromboelastometric Assays with Chicken and Human Plasma Samples

#### 5.4.1. Standardization of the Activator aPTT Clot Mean Coagulant Dose (MCD) 

The amount of subsequently described plasmatic and other additives summed to a final volume of 340 µL. Samples were composed of 260 µL of human or chicken plasmas, plus 20 µL of 200 mM CaCl_2_ and 60 µL of PBS. Crescent doses of the activator aPTT clot reagent were included as a fraction of the 60 µL of PBS. The final mixture was pipetted into a disposable cup in a computerized ROTEM four-channel system (Pentapharm, Munich, Germany) at 37 °C, and then rapidly mixed into the cup against and then away from the plastic pipette three times. The parameter evaluated was clotting time (CT, time from the start of the reaction to initial clot formation) [23]. Data were collected for one hour. Two experimental groups were considered: (Group 1) - 60 µL of PBS solution (control-group, standardized as RT), and (Group 2) - 60 µL of crescent doses of the activator aPTT clot reagent, for determination of its MCD. The MCD of the aPTT clot for CP was considered as the amount (in µL) that shortens the CT of the control group (2247 ± 319 sec) to 900 s (situated between the minimum and maximum coagulant responses). In addition, the MCD of the aPTT clot reagent for human plasma was considered as the amount that shortens the CT of the control group (874 ± 230 s) to an interval between 100 and 240 s (normal range values for the INTEM assay, according to the manufacturer’s instructions). 

#### 5.4.2. Standardization of the Effective and Mean Anticoagulant Doses (EAD and MAD, Respectively) of Bee Venom Phospholipase A_2_ on Human and Chicken Activated Plasma Samples

Briefly, 60 µL of the PBS fraction of the assay containing the MCD of the activator aPTT clot in absence or presence of crescent concentrations of bvPLA_2_ were incubated during 1 min with 260 µL of human or chicken plasmas, before the addition of 20 µL of 200 mM CaCl_2_. Two experimental groups were then considered: (Group 3) - 60 µL of PBS solution containing the MCD of the activator aPTT clot, standardized as activated samples and (Group 4) - 60 µL of PBS solution containing the MCD of the activator aPTT clot plus crescent doses of bvPLA_2_, for determining its EAD. If no change in coagulation occurred compared to normal human or chicken plasma samples, the concentrations of bvPLA_2_ were increased. However, if coagulation was not detectable, then the concentrations of bvPLA_2_ were progressively diminished until at the very least coagulation was detectable. The EAD of bvPLA_2_ was defined as the sufficient minimum dose (in ng) required to displace the CT values induced by the MCD of the activator aPTT clot in chicken or human plasmas to values within the time intervals of their related RT in 100% of assays. The MAD of bvPLA_2_, defined as the sufficient doses required to displace CT values induced by one MCD of the activator aPTT clot on CP to that corresponding to values between the mean and minimum coagulant responses (in our study, 1350 seconds), can be determined through linear regression. 

## 6. Statistical Analysis

Values of the CT parameter were monitored in seconds and expressed as mean ± SD in six independent experiments. One-way analysis of variance (ANOVA) comparisons were used, followed by Newman–Keuls post hoc analysis. Values defining the MCD of the activator aPTT clot reagent, EAD, and MAD of bvPLA_2_ were determined by means of linear regression analysis. The linear regression plots were performed using GraphPad Prism 5.0 software (San Diego, CA, USA). *p* < 0.05 was considered statistically significant.

## Figures and Tables

**Figure 1 toxins-12-00079-f001:**
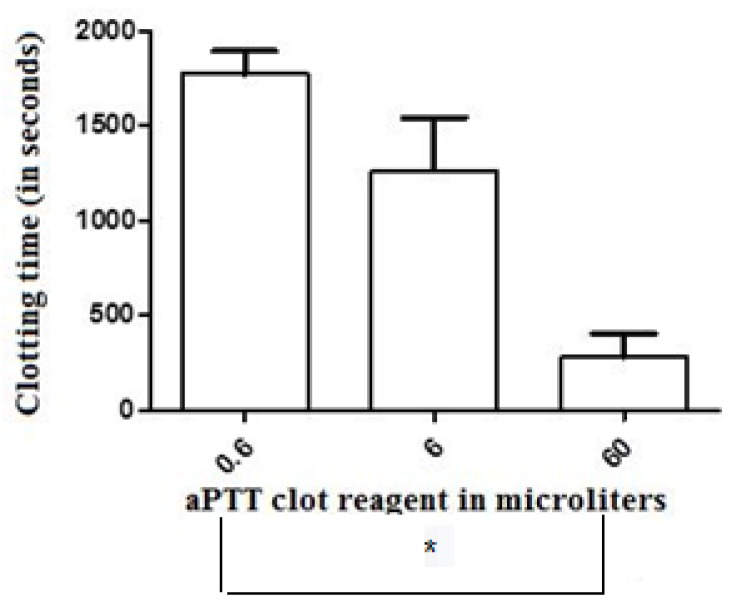
The effect of crescent doses of the activated partial thromboplastin time reagent (aPTT clot) (an ellagic acid- and phospholipid-based reagent) on the clotting time parameter of recalcified (with 20 µL of 0.2 M CaCl_2_) chicken plasma samples. Values are presented as means ± SD. * Statistically significant differences (*p* < 0.05) were observed between groups 0.6 and 60 µL, respectively (n = 6, each), treated with minor and major doses of this activator.

**Figure 2 toxins-12-00079-f002:**
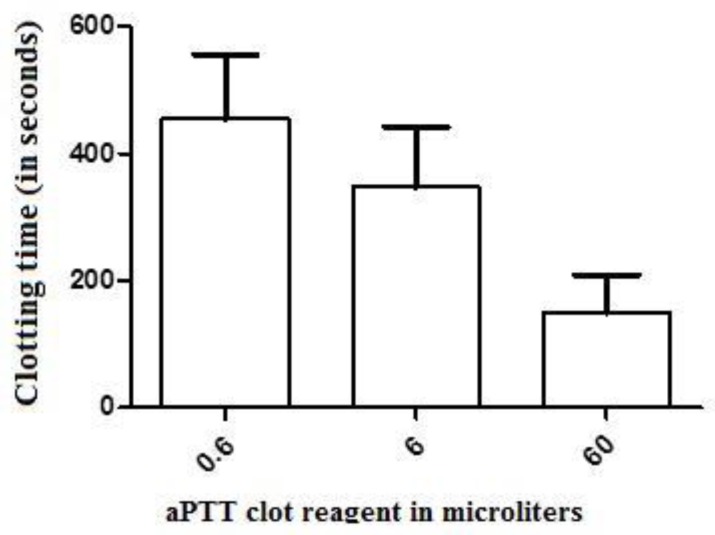
The effect of crescent doses of the aPTT clot reagent (an ellagic acid- and phospholipid-based reagent) on the clotting time parameter of recalcified (with 20 µL of 0.2 M CaCl_2_) human plasma samples. Values are presented as means ± SD. No statistically significant differences (*p* < 0.05) were observed between groups (n = 6, each) treated with minor and major doses of this activator (0.6 and 60 µL, respectively).

**Figure 3 toxins-12-00079-f003:**
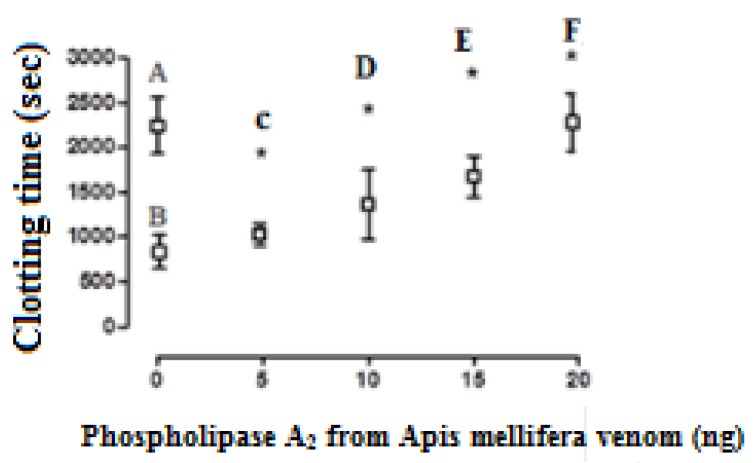
The effect of crescent doses of bee *A. mellifera* venom phospholipase A_2_ on the clotting time of chicken plasma samples recalcified (with 20 µL of 0.2 M CaCl_2_) and activated with the mean coagulant dose (MCD) of the activator aPTT clot reagent. **A**—the clotting time of the recalcified plasma samples (m = 2247 ± 319 sec, control group). **B**—recalcified plasma samples activated with a MCD of aPTT clot reagent (837 ± 185 seconds) and **C** (1032 ± 125 s), **D** (1368 ± 390 s), **E** (1679 ± 229 s), and **F** (2282 ± 319 s) are B treated with crescent doses of phospholipase A_2_. Values are presented as means ± SD. ***** Statistically significant differences (*p* < 0.05) in relation to B (n = 6 in each experimental group).

**Figure 4 toxins-12-00079-f004:**
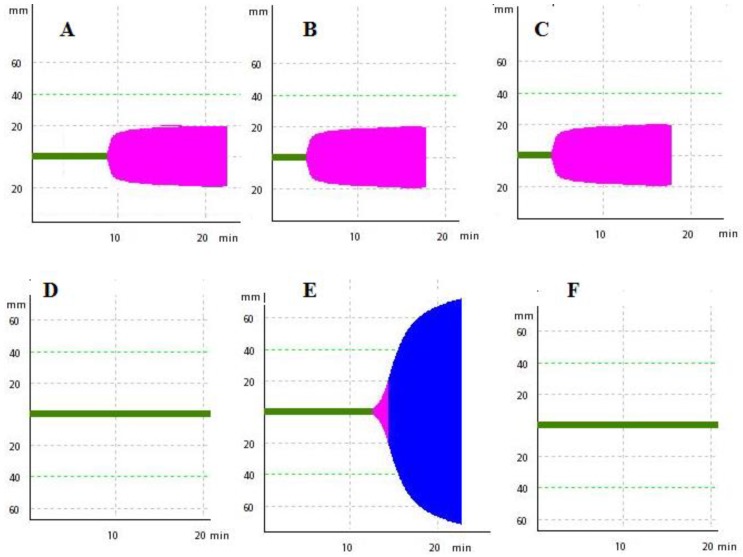
Upper, typical intrinsic pathway thromboelastometry (INTEM) profile of human plasma (260 µL) recalcified (with 20 µL of 0.2 M CaCl_2_) and treated with 60 µL of phosphate-buffered saline (PBS) solution containing: **A**—PBS solution; **B**—a mean coagulant dose (MCD) of the activator aPTT clot reagent; and **C**—**B** in the presence of 20 ng of bee *A. mellifera* venom phospholipase A_2_. Below, the typical INTEM profile of chicken plasma recalcified and treated with 60 µL of PBS solution containing: **D**—PBS solution; **E**—a MCD of the activator aPTT clot reagent; **F**—**E** in the presence of 20 ng of bee *A. mellifera* venom phospholipase A_2_, respectively.

**Figure 5 toxins-12-00079-f005:**
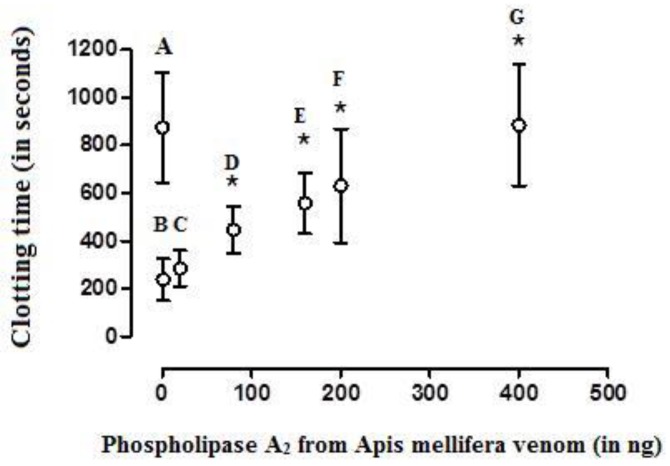
The effect of crescent doses of bee *A. mellifera* venom phospholipase A_2_ on the clotting time of recalcified (with 20 µL of 0.2 M CaCl_2_) and activated (with a mean coagulant dose (MCD) of the activator aPTT clot reagent) human plasma samples. **A**—clotting time related to recalcified plasma samples (m= 874 ± 230 sec, control group); **B**—recalcified plasma samples activated with a MCD of aPTT clot reagent (238 ± 87 sec); **C** (284 ± 75 sec), **D** (446 ± 98 sec), **E** (557 ± 127 sec), **F** (630 ± 238 sec), and **G** (884 ± 255 sec)—**B** treated with 20, 80, 160, 200, and 400 ng of bee *A. mellifera* venom phospholipase A_2_, respectively. Values are presented as means ± SD. * Statistically significant differences (*p* < 0.05) in relation to B (n = 6 each experimental group).

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
