# Peer review of "Factor XII-Deficient Chicken Plasma as a Useful Target for Screening of Pro- and Anticoagulant Animal Venom Toxins"

_toxins, 2020, doi:10.3390/toxins12020079_

Round 1
Reviewer 1 Report
Major remarks:
What could be effect of A. mellifera venom phospholipase A2 on thrombin generation using chicken plasma?
Could the authors perform turbidity assays ? And test the effect of venom on fibrin generation using chicken plasma?
What is also different between human and chicken plasma ? in term of plasmatic factors, molecules ?
It is not clear how the authors did perform clotting assays ?
The experiments should be confirmed using PRP, ie. clot retraction, fibrin formation assays
Minor remarks:
Following sentence is not clear :
Our objective was to compare the sensitivity of recalcified human and chicken 9 plasmas to bee venom phospholipase A2 (bvPLA2) anticoagulant activity, through an intrinsic 10 pathway thromboelastometry (INTEM) assay.
English of manuscript needs to be improved
Reviewer 2 Report
The manuscript entitled “Prolonged recalcification time confirms factor XII-deficient chicken plasma as a useful target for studying pro- and anticoagulant animal toxins” raises an interesting topic of venom components on the clotting of vertebrate’s blood. I enjoyed reading this well written MS. The MS presents results which are clear and have the potential to add to the literature. I have only few minor comments which I believe can improve the MS.
Lines 10, 51, 122, 202, 213 and 215 - unify the font size
Line 10 and following – unify the abbreviation for Phospholipases A2 from Apis melifera, use everywhere bvPLA2
Line 86 – Figure 1 and following, please add “values are presented as means ±SD”
Line 125-133 – I believe this paragraph is better suited for Introduction.
Lines 157 & 160 – replace “Ca2+” with “Ca2+”
Lines 205 & 206 – as bees and scorpions are arthropods this sentence is a bit unfortunate. Please rephrase it.
Round 2
Reviewer 1 Report
The authors addressed the concerns of this Reviewer.